# The Weak Spots in Contemporary Science (and How to Fix Them)

**DOI:** 10.3390/ani7120090

**Published:** 2017-11-27

**Authors:** Jelte M. Wicherts

**Affiliations:** Department of Methodology and Statistics, Tilburg University, Warandelaan 2, 5037 AB, Tilburg, The Netherlands; j.m.wicherts@uvt.nl; Tel.: +31-134-663-215

**Keywords:** reproducibility, replicability, validity, questionable research practices, meta-research

## Abstract

**Simple Summary:**

Several fraud cases, widespread failure to replicate or reproduce seminal findings, and pervasive error in the scientific literature have led to a crisis of confidence in the biomedical, behavioral, and social sciences. In this review, the author discusses some of the core findings that point at weak spots in contemporary science and considers the human factors that underlie them. He delves into the human tendencies that create errors and biases in data collection, analyses, and reporting of research results. He presents several solutions to deal with observer bias, publication bias, the researcher’s tendency to exploit degrees of freedom in their analysis of data, low statistical power, and errors in the reporting of results, with a focus on the specific challenges in animal welfare research.

**Abstract:**

In this review, the author discusses several of the weak spots in contemporary science, including scientific misconduct, the problems of post hoc hypothesizing (HARKing), outcome switching, theoretical bloopers in formulating research questions and hypotheses, selective reading of the literature, selective citing of previous results, improper blinding and other design failures, p-hacking or researchers’ tendency to analyze data in many different ways to find positive (typically significant) results, errors and biases in the reporting of results, and publication bias. The author presents some empirical results highlighting problems that lower the trustworthiness of reported results in scientific literatures, including that of animal welfare studies. Some of the underlying causes of these biases are discussed based on the notion that researchers are only human and hence are not immune to confirmation bias, hindsight bias, and minor ethical transgressions. The author discusses solutions in the form of enhanced transparency, sharing of data and materials, (post-publication) peer review, pre-registration, registered reports, improved training, reporting guidelines, replication, dealing with publication bias, alternative inferential techniques, power, and other statistical tools.

## 1. Introduction

The model depicted in Figure 1 represents a dominant model of how hypothesis-testing research in many scientific fields is (or ought to be) conducted. In this so-called Hypothetico-Deductive (H-D) model [1,2], the researcher starts with a fairly inductive (observational) phase in which he or she uses previous results, theories, and ideas from the literature to deduce a specific hypothesis. Subsequently, the researcher chooses a particular design, involving measures to be taken, treatment(s)/induction(s) to be implemented, sample(s) to be drawn, and setting(s) to be used in putting the hypothesis to the test. Given the design, the researcher now has formulated a specific prediction in the context of the study. The researcher collects the data and analyzes them. Then follows the evaluation of the evidence, which is also typically the phase at which the researcher presents the results to his or her peers through a manuscript, thesis, conference presentation, poster, or article. This contribution to the scientific literature completes the empirical cycle, leading to the next step towards “the truth”.

Looking at a very large number of empirical articles, Fanelli [3] found that the vast of majority of research articles in many fields report evidence in favor of the hypothesis. My own field (psychology, together with psychiatry) tops the list with over 90% of articles reporting positive results. Agricultural sciences and plant & animal sciences reach success rates of 83% and 77%, respectively. Around 88% of articles in biology and biochemistry offer support for the research hypothesis. But we should not become elated with joy with these success rates. There are many reasons to believe that, taken together, the stories are too good to be true [4,5].

The goal of this review is to discuss weak spots in contemporary science that create an overly positive and hence untrustworthy picture of actual effects and associations, and work towards solutions to these weak spots. I will use various animal welfare studies as examples, but discuss the problems in a generic manner. The reason is that I have become convinced that the problems in science are similar across many fields, because they are caused by the researcher’s own biases and the incentive structures that are fairly consistent across many fields. So in terms of issues of reproducibility and replicability, there is not much difference between a neuroscientist analyzing data from a ground-breaking functional MRI experiment, a medical researcher who believes she is on the verge of finding a novel cancer treatment, an economist who has found a mechanism of market collapse, a psychologist who studies how to improve intelligence, or an animal welfare researcher who is almost certain to have found the best way to keep poultry. Many of these researchers, albeit not all, work in the H-D model and will be keen to write a paper with some positive (typically significant) outcome in support of their own special breakthrough that will give them a high-impact publication, recognition by their peers, grants, and a tenured position.

Focusing on the H-D model in Figure 1 (see also [6]), I will discuss scientific misconduct, theoretical bloopers and selective readings of the literature, problems in the design of studies, issues in the analysis of data, short-cutting the cycle, and problems with the presentation (or failure to do so) of results. I will then discuss what I believe are the key psychological (human) factors that give rise to these problems, and end with a positive note on how to effectively deal with the weak spots of science in order to strengthen it.

## 2. Misconduct and Other Ways of Cutting Corners

Fraud cases, like the infamous Diederik Stapel case, [7] raise a lot of interest both within and outside academia, but are less interesting for methodological or moral debates because everyone knows it is inexcusable to fabricate or falsify data as Diederik Stapel did in over 50 of his psychology articles. What is perhaps most interesting about these cases is that they involve presentations of the data that were completely in line with the H-D model. In reality, Stapel and others who violated scientific integrity by fabricating data, did away with data collection altogether. Such a strategy clearly discards the essential step in the scientific method, but occurrences like these are quite rare; in anonymous surveys around 2% of researchers admit to using this strategy [8]. The method clearly falls outside of the scientific method, is a moral no-brainer, and often yields fabricated data with anomalous patterns that can be detected by using several statistical tools [9]. The final verdicts of misconduct involve legal procedures that need to determine who did what also fall outside of the academic sphere.

But besides fabricating data, there is another more mundane manner to cut through the empirical cycle and this strategy is called Hypothesizing after Results are Known or HARKing [10]. In one version of this strategy, the researcher collects a wealthy dataset and subsequently goes on a fishing expedition to find patterns that meet a significance threshold or appear meaningful on other grounds. Such fishing expeditions might uncover meaningful patterns, but they normally harvest old shoes that might appear impressive at first, but are fairly useless on closer inspection. Exploring data guarantees that one finds some extreme. I illustrate this problem in Figure 2, where I simulated normal white noise over a series of measurements taken over 500 days. If one looks carefully, the data show a clear and highly significant spike around *t* = 148. Now, having seen the data, it is easy to attach a narrative to this spike. Within a few moments of searching on Google (I am not making this up), I found William Shakespeare’s Sonnet no. 148, which is about love. So, there exists a poetic explanation for the spike at *t* = 148. The sonnet reads like this:
O me! what eyes hath Love put in my head,Which have no correspondence with true sight;Or, if they have, where is my judgment fled,That censures falsely what they see aright?If that be fair whereon my false eyes dote,What means the world to say it is not so?Source: The 148th Sonnet by William Shakespeare (from https://en.wikisource.org/wiki/Sonnet_148_(Shakespeare)).

The sonnet is all about how the obscured view of reality causes the poet to fall in love. Now, in this case I too fell in love with the spike at *t* = 148; it is, of course, mere chance.

This example illustrates the key problem with HARKing; the exploration selects for random occurrences in the data and a narrative is subsequently found to explain them. There is nothing wrong per se with the exploration of data, but such explorations should not be presented in the H-D model, and require a different statistical approach [11]. A problem is that researchers might feel inclined to present the results according to the H-D mold, and in fact several famous psychologists once recommended researchers to actually use this strategy of fishing to find interesting patterns in the data and to subsequently formulate hypotheses (good narratives) after the fact [12,13]. Because of the almost universal H-D mold in publishing, we cannot tell how many of the positive results in the literature are based on HARKing. Around 30% of anonymously surveyed psychologists admit to having used this practice in their work [14,15]. HARKing can take different forms. The Shakespearean example is extreme, but loosely measuring a host of outcomes to see which ones show an effect of a particular treatment and subsequently focusing on those that do in the presentation of results is statistically speaking quite similar. The same applies to measuring several covariates to see whether a given treatment is moderated by particular covariates (“interact with” in the Analyses of Variance or ANOVA framework) too could represent a form of HARKing if the hypotheses concerning these moderations (interactions) were not stipulated in advance. The explorations for extremes implicit in HARKing greatly affect false positive rates, significance levels, effect estimates, and generally, the evidence in favor of found patterns. It is almost guaranteed that the any noisy patterns found will not be replicated in novel samples of the same population.

A special case of HARKing is the switching between different outcome variables in randomized clinical trials (RCTs). Thanks to medical authorities (like the Federal Drugs Administration or FDA in the United States of America) demanding that such RCTs are pre-registered in advance, it is easy to compare the plans for what would be the primary outcomes and secondary outcomes in such RCTs with the primary outcomes and secondary outcomes reported in research articles. In doing so, many studies have uncovered that many—if not most—RCTs switch between outcomes in reporting their results [16,17,18,19,20,21,22,23,24,25,26]. Although such outcome switching could offer new insights (the workings of Viagra were uncovered as a secondary outcome), the mere presence of many potential outcomes guarantees finding at least some outcomes that show interesting results. However, the necessary statistical corrections for multiple testing are hardly used.

## 3. Theoretical Bloopers and Selective Reading

Errors in the formulation of the hypothesis are easily made, both in the inductive phase wherein earlier evidence and theoretical insights are collated (normally based on the literature) and in the deductive phase in which the ensuing general ideas (typically but not always well-developed theories) are used to formulate a specific hypothesis that is to be tested in the study. These inductive and deductive phases should follow logic, but are undoubtedly hard and require creativity, knowledge, experience, and other typically human characteristics [1]. In some fields, the study of the soundness of reasoning represents an entire sub-discipline, like in theoretical physics. But in many other fields, theoreticians are also the empiricists themselves, and it is not guaranteed that one person is well versed in both crafts. It depends on the subject matter whether the theoretical reasoning is sound, but typically peer reviewers are asked to consider this in assessing the rationale for the study. In multidisciplinary research, it is important to team up with experts in the relevant areas. Note that the reviewing might take place in earlier phases (e.g., in assessments of grant proposals) of the cycle and in the later phases (as part of the selection of articles). The earlier the assessments of theoretical soundness, the better. Improper rationales or needless repeating of earlier research could present a major waste of resources [27].

Correct referencing of the relevant literature is a related problem that is particularly vexing in increasingly large fields and is expected to become more severe due to the continuing proliferation of scientific research and the multidisciplinary nature of studies. It is common to start a research article with sentences like “Duck, Duck and Duck [1] and Mouse [2] found that Donald Duck showed massive job turnaround at a rate of around once every month in the last decade…” This is understandable, but the reader cannot be certain that the authors put in any effort to systematically collate all the available relevant evidence. In my view, we would have to make statements like “In September 2017, we searched the entire Disney database for years 2007–2016 using the search terms ‘Donald Duck’, combined with ‘work’, ‘profession’ and ‘job’, limiting our search to English articles and using our inclusion and exclusion criteria given in the flowchart in online Appendix A. We subsequently used the protocol given in online Appendix B to score the number of jobs Donald Duck has had during that period”. In short, the authors should follow guidelines, such as the PRISMA guidelines [28] in how to conduct and report a systematic review (or refer to such a rigorous review if it exists).

Without any rigorous systematic review of earlier evidence, it is quite likely that the authors overlook relevant results or insights, or that the selection of referenced results is biased. The hypothesis at hand might have been stated, refuted, or confirmed by others. Here, too, we would hope that reviewers and editors would help the authors by pointing out missed references. However, the lemming behavior that is evident in much of the bibliometric literature can be a major source of bias. Specifically, earlier (seminal) findings in research lines are typically well known, published in top journals (with high impact factors), almost always positive, and receive more citations than subsequent findings that are not novel, not published in top journals, and less positive or even completely negative [29,30,31,32,33,34,35]. Because positive, new, and well-known findings stand out, negative or other less prominent findings might well be missed in the review of the literature, leading to biased inferences required for setting up the study. So, a failure to systematically review all relevant evidence might create situations in which research has already been done, is based on less strong evidence, or would actually contradict earlier findings.

## 4. Improper Blinding and other Design Failures

The setting up of studies requires choices on the sample, induction or treatment, study setting, measurement of outcomes and control variables, and a host of other specifics. Methodologists have made this their key focus. A key design issue in the assessment of human and animal behavior, or in any diagnostic study involving human assessments and experimenters, is that expectations on the part of assessors and experimenters do not affect either what is observed or the observed behavior itself. The supposedly smart horse of Mr. von Osten [36] who was once thought to be able to do arithmetic but turned out to carefully read his master’s and his audience’s response to arithmetic questions is an excellent case that continues to be relevant today [37]. It is quite disturbing that Tuyttens et al. [38] recently found that only a minority of attendees of a major applied ethology conference was aware of the need to control for observer biases and potential experimenter expectancy effects. Such controls are prescribed in clinical trials and recommended in most textbooks in the behavioral and social sciences, but it is not entirely clear whether animal welfare researchers are sufficiently well educated to appreciate their value in designing and executing studies. Lack of blinding is quite common in animal research [39] and in the life sciences [40] and could create substantial bias in favor of expected outcomes.

Besides proper blinding, many other design aspects are crucial for the validity of the study. This includes proper controls for confounding factors via randomization and use of proper control conditions [41,42,43,44]. In addition, the chosen measures to be taken should fit the constructs of interest and should be sufficiently reliable and valid [45,46,47]. Both classic and modern psychometric theory offers many useful tools for assessing reliability of measures bearing on animal welfare and their relations to the underlying constructs of interest. Furthermore, it is important to use a sample of animals (both species and breeds), inductions (treatments), measures, and settings that allow us to make generalizable statements. The risk of highly local results that lack generalizability is high if the design of an experiment fails to consider sampling issues.

Setting up a study in the right way is crucial, and here too the room for errors and biases is large [44]. Design weaknesses such as lack of blinding and randomization pose a real threat to the validity of research, but are fortunately considered in some detail by reviewers of grants, research plans, and in final research reports. Expert reviewers would normally consider potential biases created by poor design in determining the fate of research articles, leading researchers to learn a hard (and hopefully quick) lesson in how to design better research if they want to publish their research. Yet I have a feeling that reviewers seldom take into account the possibility that even a perfectly designed study, involving good samples, proper experimental controls, blinding procedures and the like could still yield biased conclusions because of the way researchers analyze their data.

## 5. Analyzing the Data

In most research, the analyses of the collected data involve a host of decisions concerning such issues as the inclusion and exclusion criteria of research subjects and time points, dealing with outliers, handling missing data, operationalization of the outcome variable(s), use of co-variates, etc. For typical experiments in psychology, we recently identified at least 15 such analytic choices [48]. Although other types of studies and other research designs in other fields would not all involve the same choices, the complexity of research seldom creates data sets that can be analyzed in only one way. Many decisions like this in terms of the analysis of the data are defensible on methodological, statistical, or substantive grounds, leading to numerous alterative analyses of the data that are reasonable on a priori grounds [49,50]. The risk of bias is caused by researchers using these choices in a data driven manner with the goal to obtain significant or otherwise desirable outcomes. For instance, they might add a covariate, alter the inclusion and exclusion criteria, change the way the scores on the primary outcome are computed, pick another statistical estimation technique, or delete outliers in alternative ways to see whether the novel analyses did yield a desirable outcome whenever the first analysis failed to offer what the researcher had expected or hoped.

In the context of significance testing, this problem is called p-hacking [51] or significance chasing [4]. P-hacking leads to an inflation of the false positive rate to levels that far exceed the nominal significance rate (typically 5%) [52]. Genuine effects will become inflated, particularly when samples are small and underlying effects are relatively subtle [53]. Even though statisticians have attempted to develop statistical tools to detect p-hacking and to correct for its biasing effects, these effects have been hard to detect and correct after the analyses are conducted and results reported [54,55,56].

Statistically, the use of data to determine the specifics of the analysis represents overfitting, leading to exaggerated significance levels and hence a false sense of security over the robustness of the effects thus found. In many ways, the effects of p-hacking are similar to the selection of peaks as in Figure 2. The main distinction between p-hacking and HARKing concerns the hypothesis. In HARKing the hypothesis is based on the data, whereas in p-hacking the specifics of the analysis relevant for a given hypothesis are based on the data. Regardless of the role of prior hypotheses, p-hacking will result in effects becoming much weaker or even entirely absent in novel (untouched) samples of the same population, and hence in poor replicability. It is also quite likely that a p-hacked result is not robust with respect to decisions in the analyses, which leads to poor reproducibility. Steegen and colleagues using psychological data [49] and Patel and colleagues using medical observational data [57] demonstrated the sensitivity of outcomes resulting from researcher’s opportunistic use of analytic flexibility. Unfortunately, the common reluctance to share data often precludes independent assessments of the severity of bias due to p-hacking [58]. Although it is not clear how often researchers resort to p-hacking, the risk of bias due to opportunistic use of degrees of freedom in the analysis of data is large.

## 6. Reporting (or Failing to Report) the Evidence

While some fields (and journals) welcome non-significant or negative results (or even descriptive results) more readily, the de facto norm in many fields is to publish only positive outcomes [3]. Add to this the rewards for novelty, widespread focus on impact as evidenced by citations, and research grants that also demand novel contributions, and one is left with a great deal of potential bias towards finding nice, new, and (usually) significant results. Ideally, of course, the study in question provides clear answers and can be published as it is. But authors are often faced with critical reviewers and highly selective editors, who might expect (arguably unrealistically in many well intended studies) to see clear and clean results. Combined, these pressures might partially cause researchers to leave out particular results from their report, like in selective outcome reporting (or outcome switching) that is quite well-documented in the randomized trials in medicine and other fields [16,17,18,19,20,21,22,23,24,25,26,59]. Alternatively, the authors could spin the results in such a way that their outcomes do tend to appear quite good after all. This tendency to oversell the results is pervasive in the biomedical literature [60].

Or, perhaps as a last resort, a researcher could slightly misrepresent the outcomes of some numerical results if the results turned out to be disappointing. Thanks to the clearly specified reporting format for statistical test results in psychology, we have been able to document pervasive misreporting of significance test results in psychological articles [61,62]. Specifically, in around 12.5% of psychology articles that used significance testing, we found at least one non-significant result that was incorrectly presented as being significant, or vice versa. Discrepancies like these could be the result of honest error, but the practice has been earmarked as one of many questionable research practices in anonymous surveys among psychologists, [14] and the high prevalence of such inconsistencies does suggest that it is quite common. Other fields show somewhat lower, albeit still quite high prevalences of statistical reporting inconsistencies [63,64]. Reporting can be erroneous or incomplete in multiple ways [65].

Moreover, a researcher confronted with less desirable (typically non-significant) results, could also just decide not to report the study altogether. The problem is often called the file drawer problem, but is not altogether clear whether the missing studies are indeed kept in a digital or physical archive wherefrom they can one day be readily retrieved to be eventually disclosed and hence used in publications or research syntheses. Given the poor documentation of materials and data as evidenced by surveys in many fields [66,67], I fear that less desirable results often just end up in the digital or physical thrash can. Either way, the file drawer problem or publication bias is widely considered problematic for scientific progress [68], because it creates a selection of positive results into the literature. The literature on animal welfare might well be affected by such publication bias [69]. Publication bias negatively affects the validity results of (systematic) literature reviews and meta-analyses, and generally paints a picture that is too good to be true. Coupled with p-hacking and other problems in science, publication bias can create a literature filled with many false and inflated research results [4,5]. Because publication bias is often caused by selection on significance [70,71], fields that involve weaker effects and smaller samples (key ingredients for non-significance even if an effect is genuine) are particularly vulnerable to it.

## 7. Researchers are Only Human

I have identified several of the weak spots in hypothesis testing research as it is done in many scientific fields. These include—but are certainly not limited to—misconduct, HARKing, poor theorizing, incomplete reviewing of earlier results, poor design, publication bias, and biases caused by the way researchers analyze their data and report their results. These problems occur despite much apparent progress in many fields and notwithstanding the good intentions of the vast majority of researchers. There are many different ways of assessing the underlying causes of problems in science, including assessments made using statistical, sociological, economic, and even evolutionary analyses [4,6,72,73,74,75,76]. My own background in psychology lets me focus on the psychological factors in creating these issues and so I am inclined to assess these problems in terms of the human factor, i.e., from the notion that researchers are only human. Human factors that create biases include the poor statistical intuitions that lead to an underappreciation of randomness [77,78], confirmation biases leading us to hold a double standard when it comes to assessing evidence for and against our expectations [79,80,81,82], and hindsight bias or the unjustified feeling that “we knew it all along” even we did not (as is evident in HARKing [83]).

Researchers appear to want to do it well. In a large survey [84] of over 3000 researchers funded by the National Institutes of Health (NIH), Anderson and colleagues asked whether they subscribed to the good norms of science, like the focus on rigor, disinterestedness, objectivity, and openness. The good news was that the vast majority of researchers (over 90% at both early career and mid-career levels) indicated that they subscribed to these norms. But Anderson et al. [84] subsequently asked whether the researchers also behaved according to the norms and whether they felt that their typical (scientific) peer behaved according to these norms. And that result offered the bad news: around 30% of researchers indicated to not always behave in line with the norms of good science. Worse still, the surveyed researchers overwhelmingly (over 60% of them) indicated that they felt that their peers were mostly in it for the money and would value quantity over quality, secrecy over sharing, and self-promotion over a desire for knowledge. Even though these common beliefs concerning the unscientific norms of others might be too negative, it does create a problem when there is competition between researchers for scarce resources and academic positions. Even if one wants to do what is best for science, one might sometimes feel inclined to choose what is best for one’s career. These goals could be inconsistent and raise moral dilemmas [85].

Even honest people sometimes act slightly dishonestly in particular circumstances [86]. One of my favorite experiments in the growing field of moral decision-making let its normal adult participants throw a die to determine how much money they would take home after leaving the laboratory [87]. Each eye of the die was worth one dollar, with a one resulting in the lowest pay and a six resulting in the highest pay of six dollars. The participants were instructed in a closed test cubicle via the computer and threw the die in such a way that no one could ever determine what they actually threw, namely under a cup with a small hole in the bottom through which they alone could see the outcome before reporting it. Now economists would expect that in this uncontrollable environment anyone would maximize their profit by reporting sixes, but this is not what the experiment showed. When the researchers let the participants throw the die only once, the distribution of reported throws did not deviate from the (uniform) distribution that one would expect under honestly reported throws of the die. So, there were no more reported sixes than expected by chance. The explanation for this is that participants see themselves as honest and they would like to continue to look at themselves in the mirror as such an honest person. Lying for such a small profit simply does not compensate for the guilt associated with violating one’s honest self-image. This suggests that we are equipped with a moral compass. However, results changed dramatically when the researchers let the participants (again in the same uncontrollable environment) throw the die not once but multiple times. According to the instruction, the first throw would determine the pay, but the distribution of reported outcomes strongly suggested that, instead, the participants reported the highest of three throws of the die (which is the typical number of throws when one is asked to throw multiple times). It is easy to envisage how this works: the first throw is a one, but the second throw is a five. One thinks (rightly!) that the result could have been a five in the first round, and subsequently reports the five without any guilt. Thus, this study and others like it showed that honest participants tend to lie for profit in some circumstances.

The researchers interpret this result as showing that participants see their moral transgressions as less problematic when they are offered a justification for them. Further results in this paradigm has shown that honest people are inclined to lie (a little!) when they feel that others would do it as well, when the situation is ambiguous, and when the lie is seen as inconsequential and harmless [88]. Although this research has not yet been replicated among researchers conducting analysis of their data, these circumstances do come across as similar to a researcher who is doing precisely that. The outcome of a statistical analysis is dependent on a host of random factors and can be done multiple times (quite quickly with modern computers), and hence running analyses are much like throwing a die. The analytic results are heavily incentivized (require significance), we might believe that others would do the same thing (see Anderson et al.’s survey [84]), and the effects cannot possibly be bad. Confronted with a non-significant result, the conversation could run as follows:
We started out with an elegant hypothesis that could improve the world.Surely there must be something wrong with that first analysis.That second analysis could have been the first analysis, so why wouldn’t we report that second one?

This psychological mechanism here could be quite potent. Combined with poor statistical intuitions (statistics is hard and often counterintuitive) and several other cognitive biases, this explains why researchers resort to p-hacking, potentially without being much aware of it. I believe that researchers are human when they p-hack and so there is no need to point moral fingers. They might genuinely believe that the patterns found in the data are given to them by the data instead of being the result of their own repeated testing. But the statistical results of repeated analysis is akin to finding accidental spikes like we did in Figure 1; data-driven analyses like this yield results that are likely to be inflated or simply false. Fortunately, this explanation of why we resort to p-hacking does offer some way out in the form of some solutions.

## 8. Solutions

We need to deal with these human factors in science, and luckily several solutions have been proposed [6,85,89,90]. Most proposed solutions are based on the notion that researchers overwhelmingly want to do it well. The hope is that different scientific stakeholders (editors, academic institutions, science publishers, professional societies, and grant organizations) will also pay attention and implement policies to reward responsible research practices that strengthen science. Researchers can certainly change their ways, but if these stakeholders ignore the scientific benefits and continue to reward only novel and positive results, systematic change will be a distant hope.

### 8.1. Transparency and Post-Publication Peer Review

Independent assessments of research design and results by scientific peers are considered to represent a crucial step in awarding research grants and in publishing the research. Peer review could also play an important role in assessing and countering biases and other problems in science. If research materials (e.g., protocols, questionnaires, physical specimens, treatment specifics) and research data (both raw data and processed data) including useful computer scripts or syntaxes were made openly available alongside research reports, the chances of being caught HARKing and conducting misconduct could be greatly enhanced [91]. In addition, it would be easier to assess potential biases through independent re-analyses of the data (which would strengthen reproducibility) and to replicate the research in novel samples (which would strengthen replicability) [90]. Hence, having a more open research culture could help in countering biases [92]. Journal publishers could demand more transparency from their authors [93] or could incentivize data sharing and other open practices by handing out badges, which appear to work quite well in promoting rigorous practices [94]. Data repositories like Figshare, Dataverse, Dryad, and (online) platforms like the Open Science Framework (https://osf.io) or the Jackson Laboratory (https://www.jax.org) could greatly help in sharing these materials, computer syntaxes, physical specimens, and data. Similarly, the Resource Identification Initiative helps researchers to identify key research materials (see: https://scicrunch.org/resources). With these technical capabilities, which are typically offered for free, not sharing data or (identifiers for) key research materials will hopefully become an anachronism of the time when data and materials were still almost entirely put in heavy boxes or unavailable to begin with. Sharing of data and materials requires considerations of user rights (the data collectors would want to be the first to harvest the fruit of their labors) and privacy issues for human data, but these issues can be dealt with [95,96,97].

Ideally, reviewers would consider transparency in their assessments of the validity of research before publication. However, peer reviewers do not always spot errors in submitted work [98]. Additionally, post-publication review via online platforms (either at the publisher site or elsewhere) could help later corrections for biases in research articles. Combined with open data (allowing scrutiny), use of preprints, and open access publishing (having a broader readership), more extensive review before and after publication could strengthen the key scientific mechanism of self-correction. The core idea is that transparency empowers scrutiny, which is key to assessing scientific trustworthiness.

### 8.2. Pre-Registration and Registered Reports

An effective antidote to HARKing, the opportunistic use of flexibility in the analyses (p-hacking), and overly positive reporting is pre-registration [67,97,99]. In pre-registering their study, the researchers explicate the hypotheses, design choices, and analytic approaches before starting the data collection. This so-called pre-registration should be openly published, time-stamped (and should not be changed), and sufficiently detailed to ascertain that the study was actually done in line with the hypothesis testing model in Figure 1 and free from biases due to how researchers analyze the data and report the results. It is important that the pre-registration is sufficiently specific to disallow biases to reemerge during the entire research project [48], including not only the plans for analyzing the data, but also specific design characteristics and specifications of how the data will be collected. For instance, a failure to specify precisely how outliers will be identified and handled in the pre-registration would still allow the researcher to try different approaches in an ad hoc and goal-directed manner [100], thereby creating bias. Similarly, loosely described data collection procedures could introduce biases such as those caused by decisions to discard data that appear not to fit the predictions. Moreover, common failures to follow protocols as part of registrations of randomized clinical trials in the biomedical sciences [16,17,18,19,20,21,22,23,24,25,26] highlighted the need to have reviewers of later articles compare the registrations to the later reports.

An interesting new development (with old roots [99]) is the registered report [101]. In this publishing format, the review concerns the registration rather than the final research article, allowing reviewers to catch potential errors and oversights in the formulation of the hypotheses as well as to make recommendations to improve the design at precisely the right moment. If reviewers are happy with the hypotheses, design, and analyses, the researchers get an in principle acceptance of their later article and they can set out to collect their data and analyze them as stipulated in advance. If the data collection goes according to plan, the article is published regardless of whether the results are positive or not. Hence, registered reports focus on the a priori methodological and substantive quality of the study rather than on the niceness of the results. Hence registered reports also help counter publication bias. Although traditional grant review also involves peer review of research plans, plans in grant proposals are often not sufficiently detailed to reveal biases during the actual research process and in the later (non-) publication of results. Therefore, it would be much more efficient to use registered reports in a format in which the grant review and publication decisions are combined and focused on the specific registration of a study [102].

It is important to note that any pre-registered study could still offer novel results based on explorations of the data. However, it is key to report these explorations as being different from the key confirmatory hypotheses that the study was set out to test. Obviously, such explorations could identify interesting and relevant patterns in the data, but these should be further studied in novel samples in a direct replication to rigorously be tested empirically and be trusted as being more than a chance finding.

### 8.3. Improved Training and Reporting Guidelines

There is widespread agreement among meta-researchers, methodologists, and statisticians on the potential problems of selective readings of the literature, p-hacking, improper design, publication bias, and inaccurate reporting of results. Awareness of these issues and knowledge of solutions could certainly help in countering these problems, and education of young (and perhaps also more experienced) researchers could be improved. Biases in contemporary research should become a standard part of the curriculum. Better training could help improve the design of studies and the manner in which the data are collected, analyzed, and reported.

Methodologists, statisticians, and meta-researchers have also made extensive recommendations on the reporting of methods and results in research articles. Following reporting guidelines such as the ARRIVE guidelines [103] for reporting animal studies, the STROBE guidelines [104] for reporting observational research, and the CONSORT guidelines [105,106] for randomized trials could all help in improving the way science is reported. No single study is perfect, and so following of these guidelines allows much better assessments of the risk of bias and thereby of the validity of the results by reviewers, editors, researchers, and other stakeholders. Such risks of bias could later be assessed in systematic reviews or meta-analyses to get a better picture on phenomena which is currently often too positive and hence untrustworthy. With the guidelines on reporting also come a responsibility on the part of researchers to be open about the methodological limitations of studies and a responsibility on the part of editors and reviewers to not criticise research reports that discuss these limitations too harshly. Several studies have highlighted that the implementation of reporting guidelines is not always as it should be [107,108,109], and so it is crucial that researchers, reviewers, editors, and publishers work together to ensure that research is reported well.

### 8.4. Replication and Dealing with Publication Bias

Independent replication of previous results is generally seen as the litmus test for any finding, and should be valued more and conducted more widely. Many journals continue to focus on novelty and might be less willing to publish replication studies, particularly when the results contradict earlier findings. Grant-giving organizations seldom fund replication research (although the Netherlands Organization for Scientific Research (NWO) has launched a replication grant program last year while other funders including the National Science Foundation and the Laura and John Arnold foundation have encouraged or funded replication research), and this creates a suboptimal corrective mechanism in many scientific fields. Fortunately, an increasing number of journals like PLoS ONE is now no longer using “novelty” or “importance” as publication criteria. An increased focus on methodological rigor hopefully helps to counter the biases against findings which are not considered new or positive that have long set back corrections of false or otherwise inflated results in many scientific literatures. Publication bias could be countered by publishing all relevant results [110], whereby relevance is a prior characteristic of the study rather than based on its results. Besides journals becoming more welcoming to so-called negative or null results, the increasing posting of pre-prints could help disseminate relevant results in a quicker way and make them subject to fewer biases like those that have been identified in the traditional publication formats [111].

Much discussion in psychology has revolved around the feasibility and meaning of direct replications of earlier studies. The recent large-scale reproducibility project in psychology attempted to replicate the results of 100 earlier studies reported in top journals in the field [112]. At first sight, the results might appear disheartening, with the majority of replications yielding much smaller effects than the original studies. Only 36% of the replications yielded significant results, compared to 97% in the original studies. However, much statistical debate highlighted that this result does not mean that the effects are not present in replication studies that showed no significant effects [113,114,115,116]. Replication studies could yield Type II errors or tap on effects that are genuinely different for methodological or substantive reasons (e.g., differences in contexts, treatments, and participants). Nevertheless, replications are crucial to determining the generalizability of results, and should guide future research efforts towards understanding when an effect or association is stronger or weaker. If all relevant studies (replications) on a given topic are available and free from biases due to p-hacking, a meta-analysis could be very valuable for summarizing the evidence and considering the reasons why some studies show larger effects or associations than other studies do. Meta-analysis could be very helpful in finding meaningful patterns and allow one to study moderation of effects and associations by important substantive or methodological factors. The focus on meta-analyses of multiple similar studies and effect sizes also diminishes the need to base claims on single studies.

### 8.5. Inferential Techniques, Power, and other Statistical Tools

For hypothesis testing studies, the use of null hypothesis significance testing (NHST) is clearly the dominant statistical approach. There is widespread debate on the pros and cons of NHST [117], including many discussions on alternative inferential techniques including the use of Bayesian approaches [118], estimation and confidence intervals [119], and more stringent nominal significance levels [120]. There is little doubt about the widespread misinterpretations of NHST [121,122], and it is crucial to always consider whether NHST is the most appropriate inferential approach for a given research question. None of the alternative inferential techniques is immune to biases but could certainly yield diverging inferences. Therefore, it is valuable to consider the same data from different (pre-registered) angles in the hope that they provide a consistent inference on what these data mean for the hypothesis at hand.

Many of the issues of significance testing are caused by low power [4,5,53,123]. When using NHST, is crucial to formally consider the statistical power, which is a function of the underlying effect, the nominal significance level, and the sample size. Besides using sufficiently large sample sizes (often impeded by practical constraints), there are alternative ways to increase the power, such as use of statistical controls, repeated measures designs, and a host of statistical techniques that could offer improved power. For instance, lowering the measurement error and other sources of random error during the data collection are also useful for increasing the chances of finding a significant effect. Moreover, structural equation models with latent variables could offer higher power than traditional Multivariate Analysis of Variance (MANOVA) in analyzing experimental data that involve multiple outcome measures that have a bearing on the same underlying latent factors. For instance, suppose a certain treatment attempts to lower stress levels among pigs and measures stress levels by collecting data on the heart rate variability, (blinded) observations of emotional states, and measures of stress-related vocalizations. Because we expect the covariation of these three measures to be caused by a single underlying stress factor, we could run a one-factor model and test for a mean group difference between treatment conditions on this factor. Such a latent variable structural equation model would provide better power to detect effects than multivariate or univariate analyses of variance (MANOVA or ANOVAs) because it takes into account measurement errors and tests for a single latent effect instead of multiple effects [124]. These best practices of using controls, larger samples, accurate measurement, and modelling approaches are valuable for any inferential technique because they heighten the signal to noise ratio.

Furthermore, several statistical techniques directly deal with the notion of over-fitting that is caused by researcher’s (understandable) inclination to analyze the data in different ways to find interesting (and typically significant) results. These statistical techniques include re-sampling methods, and a host of hold-out sample methods in which part of the data is used to look for patterns and another part is kept aside to verify the robustness of these found patterns in fresh new samples [125]. Such techniques can be very valuable in many research settings, because they allow explorations in so-called training data set and subsequent verification of these explorations in a hold-out sample. Such techniques require sufficiently large samples, but provide efficient and statistically sound ways to both explore and confirm findings in a data set [126] without the need to perform a direct replication in a new sample.

## 9. Discussion

In this review, I discussed several of the weak spots in contemporary science, including scientific misconduct, the problems of post hoc hypothesizing (HARKing), outcome switching, theoretical bloopers in formulating research questions and hypotheses, selective reading of the literature, selective citing of previous results, improper blinding and other design failures, p-hacking or researchers’ tendency to analyze data in many different ways to find positive (typically significant results), errors and biases in the reporting of results, and publication bias. I presented empirical results and arguments in favor of the (potential) biases that these problems could create for the trustworthiness of reported results in the scientific literatures, and discussed some of the underlying causes of these biases based on the notion that researchers are only human. I proposed several solutions in the form of enhanced transparency and (post-publication) peer review, pre-registration, registered reports, improved training, reporting guidelines, replication, dealing with publication bias, alternative inferential techniques, power, and other statistical tools. The literature on these problems is very large and growing rapidly [127], and so my review of problems is not exhaustive. Rather my review presents my assessments as a meta-researcher of the key weak spots in contemporary science as it is practiced in many fields, including that of animal welfare research.

I focused on the hypothetico-deductive model, but science certainly does not always follow this model and some have argued against it [128]. Many studies are in fact much more descriptive (exploratory), and much research could benefit from using other approaches aside hypothesis testing in the H-D model [128,129]. Also, much of my discussion of problems in science revolved around inferential techniques like the null hypothesis significance tests that are dominant in many fields. Many of the problems would apply also to Bayesian or more descriptive approaches, but there can be little doubt that the mindless use of inferential techniques is problematic and that alternative inferential techniques could help in analyzing and evaluating research results.

I deliberately approached the problems in contemporary science in a general way, because the issues are quite similar across different fields. Yet most of the empirical data on the severity of the problems in the emerging field of meta-research is focused on the medical and behavioral sciences. Future research should address whether the same problems are apparent in animal welfare research. Such an effort is valuable because it could raise awareness of the most vexing issues among working researchers in the field and other stakeholders. Moreover, documenting problems could point at the most immediate targets of interventions. Such interventions could be tested empirically using randomized designs that would require extensive collaborations with for instance journal publishers and editors, or by documenting improvements over time as policies or research practices improve. Improved methods for designing, conducting, analyzing, and reporting studies will eventually help promote scientific research that can benefit society (including animal welfare).

## 10. Conclusions

There is increasing awareness among many researchers that contemporary science is not always functioning optimally [130]. In this review, I discussed several of the core problems that lower reproducibility and replicability of research results. Meta-researchers have amassed considerable evidence highlighting the severity of these problems in the biomedical and behavioral sciences, and there is little reason to expect that such biases would be less severe in other scientific fields that rely on hypothesis testing. Luckily, several solutions exist that have the potential to improve reproducibility and replicability in many fields, thereby improving scientific progress.

## Figures and Tables

**Figure 1 animals-07-00090-f001:**
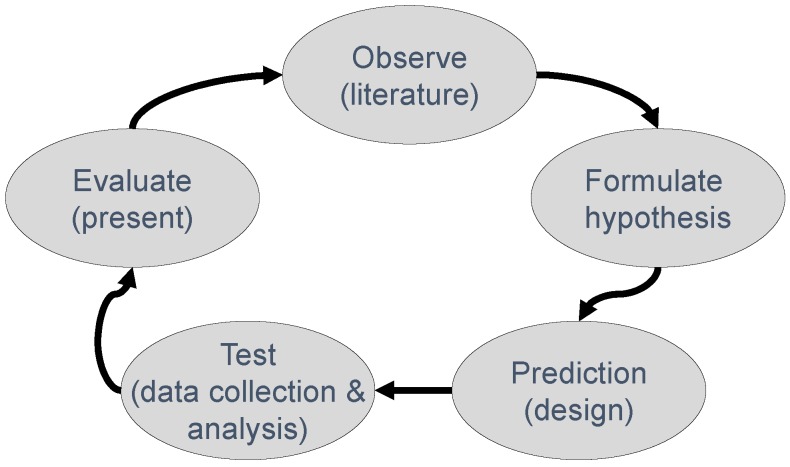
The hypothetico-deductive cycle commonly used in hypothesis testing.

**Figure 2 animals-07-00090-f002:**
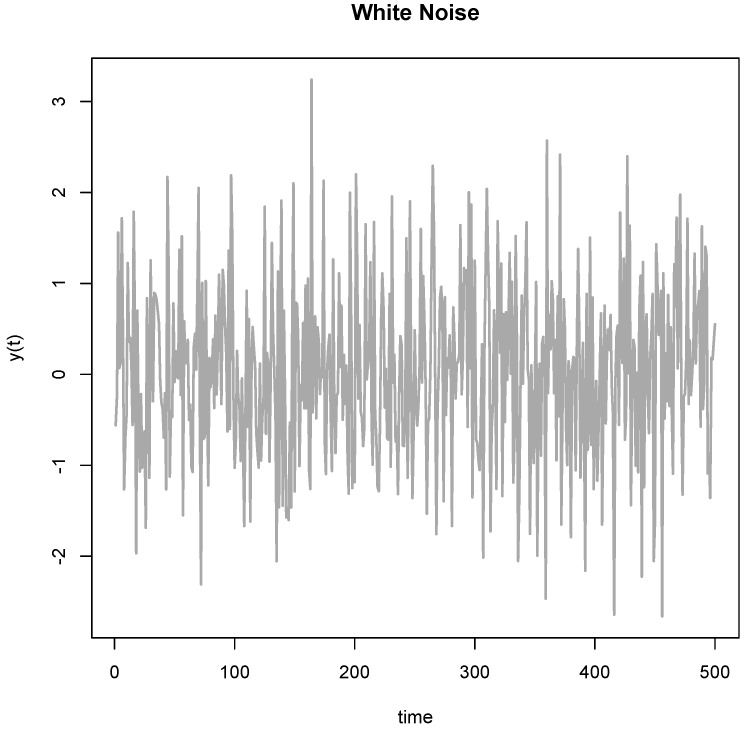
Simulated data highlighting an interesting spike around *t* = 148 explained by Shakespeare’s 148th Sonnet to illustrate the dangers of post hoc hypothesizing (HARKing). y(t) represents the hypothetical outcome variable.

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
