# Peer review of "The Weak Spots in Contemporary Science (and How to Fix Them)"

_animals, 2017, doi:10.3390/ani7120090_

Round 1
Reviewer 1 Report
As a narrative review article built on a conference presentation, the role of the peer reviewer is limited. Since what is being offered is an opinion based on the author's reading and experience, there is little to be said except either "I agree" or "I disagree". In this case, I agree, which clearly biases my assessment of the work - if the author agrees with me, obviously they have well expressed, logical and clear arguments.
I have a couple of thoughts. I think early on hypotheco deductive is expressed as HD, later H-D ... so better to be consistent
Secondly (and this is hobby-horsing on my part) I think the case that research is broken is pretty well established. What would be interesting is a deeper discussion of the evaluation of the measures which have been proposed to address this - do we need to go as far as randomised trials of publishing interventions eg to increase reporting of randomisation in animal studies, or would quality improvement strategies (where the comparison is usually before - and - after, in short plan-do-study act cycles) be good enough for many of our purposes?
Author Response
I would like to thank the reviewer for these positive comments. I have made the H-D term shorthand consistent and added a discussion of the interesting point regarding the use of interventions to study improvements in how we conduct and report our studies.
Reviewer 2 Report
This review focuses on the cause and some solutions to address weaknesses in the the way we currently conduct and disseminate research. Overall, this manuscript presents content that is of interest to a broad scientific audience.
Minor comments (not including minor spell/grammar checks):
Line 168-169: PRISMA (http://www.prisma-statement.org/Default.aspx) provides guidelines for transparent reporting of systematic reviews and meta-analysis. It might be worth citing this when discussing the need for authors to follow guidelines like these.
Line: 183 (section 4): This is an important section regarding study design and tools that should be utilized to minimize bias. Relevant to animal experiments, a review (http://journals.plos.org/plosbiology/article?id=10.1371/journal.pbio.1001756) of author adherence to the ARRIVE guidelines 2 years after some journals adopted them reviewed that not all guidelines were being reported (relevant for this section were low rates of blinding, randomization, and sample size calculations). This shortfall might be useful to discuss, particular as these guidelines are discussed again as a solution in section 8.3.
Line 213-216: This sentence is true the other way around. While likely written this way to transition from study design to study analysis, it might be worth explicitly stating. If a study, for example lacks blinding and randomization, even a good (pre-registered) analysis is prone to bias. This is hinted at in lines 408-411, but in reference to analysis (outliers), not study design considerations. It might strengthen the pre-registration and Registered Reports argument to comment how these specifically apply to design as well as analysis decisions.
Line 278-281: I (among many researchers) strongly agree that the file drawer problem is a large concern with trying to understand the current evidence. The size of course is unknown as the author suggestions, however there have been some attempts at trying to estimate it (e.g. http://science.sciencemag.org/content/345/6203/1502.long). I wonder if the author sees pre-prints as a mechanism towards allowing the null/less desirable results to be communicated?
Line 380: Research materials are more than just the digital contents used in the research, but also physical specimens (e.g rodent models purchased from suppliers). Efforts such as the Resource Identification Initiative to identify these key materials through an RRID (https://scicrunch.org/resources) aim to do this for the physical specimens that are described. Additionally, there are efforts to encourage deposition of physical specimens into repositories for reuse (e.g. donating a mouse strain to a recognized and supported repository to share with the research community (i.e. The Jackson Laboratory)).
Line 401: In section 8.2 on pre-registration, the author might consider describing how pre-registration (and Registered Reports) differ from the traditional grant review mechanisms. A new approach is also being piloted to combine grant review and the Registered Reports format (https://academic.oup.com/ntr/article/19/7/773/3106460 and http://blogs.plos.org/everyone/2017/09/26/registered-reports-with-ctf/ ).
Line 453: There are indeed few grant giving organizations that fund replication research. In addition to NWO, the NSF has recently encouraged grants to be awarded for replication research (https://www.nsf.gov/pubs/2016/nsf16137/nsf16137.jsp). Private foundations (e.g. The Laura and John Arnold Foundation) have also funded replication research (e.g. Reproducibility Projects in Psychology and Cancer Biology).
Author Response
I would like to thank the reviewer for his/her positive comments and excellent suggestions. I have revised the manuscript to deal with issues raised and suggestions made by this reviewer.
Line 183 and section 8.3. I now refer to and discuss the PLOS BIO article (and similar works) on the adherence to reporting guidelines.
Lines 213-216 and 408-411. I added a few sentences to clarify that bias can be introduced in all phases of a study, including the design, data collection, analysis, and reporting.
Lines 278-281. I added two references that provided direct evidence of the file drawer problem and discussed (in a later section) the potential of preprints in disseminating research more broadly and effectively.
Line 380. I added physical specimens to the discussion of transparency and now refer to both RRID and the Jackson laboratory.
Line 401. Again an excellent point that I now included with a reference to the OUP article.
Line 453. I added the two additional funders that either fund or encourage replication research.